# Molecular Analysis of *ZNF71 KRAB* in Non-Small-Cell Lung Cancer

**DOI:** 10.3390/ijms22073752

**Published:** 2021-04-04

**Authors:** Qing Ye, Rehab Mohamed, Duaa Dakhlallah, Marieta Gencheva, Gangqing Hu, Martin C. Pearce, Siva Kumar Kolluri, Clay B. Marsh, Timothy D. Eubank, Alexey V. Ivanov, Nancy Lan Guo

**Affiliations:** 1WVU Cancer Institute, West Virginia University, Morgantown, WV 26506, USA; qiye@mix.wvu.edu (Q.Y.); romohamed@mix.wvu.edu (R.M.); duaa.dakhlallah@hsc.wvu.edu (D.D.); michael.hu@hsc.wvu.edu (G.H.); tdeubank@hsc.wvu.edu (T.D.E.); aivanov@hsc.wvu.edu (A.V.I.); 2Lane Department of Computer Science and Electrical Engineering, West Virginia University, Morgantown, WV 26506, USA; 3Department of Microbiology, Immunology & Cell Biology, West Virginia University, Morgantown, WV 26506, USA; mgencheva@hsc.wvu.edu; 4Institute of Global Health and Human Ecology, School of Sciences & Engineering, The American University of Cairo, New Cairo 11835, Egypt; 5Cancer Research Laboratory, Department of Environmental and Molecular Toxicology, Oregon State University, Corvallis, OR 97331, USA; pearcem@oregonstate.edu (M.C.P.); Siva.Kolluri@oregonstate.edu (S.K.K.); 6Department of Medicine, West Virginia University, Morgantown, WV 26506, USA; cbmarsh@hsc.wvu.edu; 7Department of Biochemistry, West Virginia University, Morgantown, WV 26506, USA; 8Department of Occupational and Environmental Health Sciences, West Virginia University, Morgantown, WV 26506, USA

**Keywords:** KRAB isoform, zinc finger protein, EMT, prognosis, chemoresponse, proliferation, CRISPR-Cas9, RNAi

## Abstract

Our previous study found that zinc finger protein 71 (ZNF71) mRNA expression was associated with chemosensitivity and its protein expression was prognostic of non-small-cell lung cancer (NSCLC). The Krüppel associated box (KRAB) transcriptional repression domain is commonly present in human zinc finger proteins, which are linked to imprinting, silencing of repetitive elements, proliferation, apoptosis, and cancer. This study revealed that *ZNF71 KRAB* had a significantly higher expression than the *ZNF71 KRAB*-less isoform in NSCLC tumors (*n* = 197) and cell lines (*n* = 117). Patients with higher *ZNF71 KRAB* expression had a significantly worse survival outcome than patients with lower *ZNF71 KRAB* expression (log-rank *p* = 0.04; hazard ratio (HR): 1.686 [1.026, 2.771]), whereas *ZNF71* overall and *KRAB*-less expression levels were not prognostic in the same patient cohort. *ZNF71 KRAB* expression was associated with epithelial-to-mesenchymal transition (EMT) in both patient tumors and cell lines. *ZNF71 KRAB* was overexpressed in NSCLC cell lines resistant to docetaxel and paclitaxel treatment compared to chemo-sensitive cell lines, consistent with its association with poor prognosis in patients. Therefore, *ZNF71 KRAB* isoform is a more effective prognostic factor than *ZNF71* overall and *KRAB*-less expression for NSCLC. Functional analysis using CRISPR-Cas9 and RNA interference (RNAi) screening data indicated that a knockdown/knockout of ZNF71 did not significantly affect NSCLC cell proliferation in vitro.

## 1. Introduction

Lung cancer is the second most common cancer and remains the leading cause of cancer-related mortality in the U.S. Non-small-cell lung cancer (NSCLC) accounts for 84% of lung cancer cases. Major histological subtypes of NSCLC include lung adenocarcinoma, squamous cell carcinoma, and large cell carcinoma. In 2020, an estimated 228,820 adults (116,300 men and 112,520 women) in the US were diagnosed with lung cancer [1]. About 30% to 55% of the patients with NSCLC develop recurrence and die of the disease within 5 years of the surgical removal of their tumors [2]. According to the current practice guidelines, NSCLC patients with stage 2 and above receive chemotherapy, with additional radiation for stage 3A patients [3]. While adjuvant chemotherapy of stage 2 and stage 3 disease has resulted in 10–15% increase in overall survival [4], the prognosis for early-stage NSCLC remains poor [5], indicating that some patients may not benefit from it. To date, physicians do not have a precise tool to identify patients with resectable NSCLC that are likely to develop tumor recurrence or metastasis. Our previous study developed a 7-gene assay for NSCLC prognosis and prediction of chemotherapeutic benefits [6]. The ability of this gene assay to identify those at a high risk for recurrence or metastasis would potentially inform selection of specific adjuvant chemotherapy for these patients.

Among this 7-gene signature, mRNA expression of zinc finger protein 71 (*ZNF71*) was positively associated with survival in patients who received cisplatin, carboplatin, and Taxol in the studied cohorts, indicating its association with chemosensitivity [6]. Although *ZNF71* mRNA expression was not associated with NSCLC survival in the overall patient cohorts analyzed in qRT-PCR, higher ZNF71 protein expression quantified with AQUA was associated with a more favorable survival outcome in two separate NSCLC cohorts (*n* = 291) using tissue microarrays (TMA) [6]. Zinc finger proteins (ZNFs) are involved in DNA repair, degradation of proteins, signal transductions, migration of cells, regulation of apoptosis, lipid binding, and transcription regulation [7,8]. The Krüppel associated box (KRAB) is a transcriptional repression domain and is commonly present in human zinc finger protein-based transcription factors, i.e., KRAB zinc finger proteins (KRAB-ZFPs) [9,10,11]. Transcriptional repression mediated by KRAB-ZFPs is linked to cell proliferation, differentiation, apoptosis, and cancer [12]. ZNF71 (EZFIT) was first identified by Mataki et al. [13] as a ZFP induced by tumor necrosis factor α (TNF-α) in human umbilical vein endothelial cells. Single-nucleotide polymorphisms of ZNF71 were found to be linked with total serum IgE in Korean asthmatics in a genome-wide association study [14]. To the best of our knowledge, there has been no report on molecular analysis of ZNF71 KRAB isoform in cancer.

Epithelial-to-mesenchymal transition (EMT) is a highly dynamic process in which epithelial cells can convert to a mesenchymal phenotype. EMT is also reversible by the mesenchymal-to-epithelial transition (MET). Emerging evidence reveals the involvement of EMT in tumor progression, metastasis, and resistance to cancer treatment [15,16,17,18]. However, the involvement of EMT in cancer patient outcomes remains controversial [19]. Recent studies have evaluated EMT as well as stromal and immune infiltration in tumors using transcriptional profiles [19,20,21]. This study sought to investigate the expression of *ZNF71 KRAB* and *KRAB*-less isoforms in NSCLC tumors and cell lines and their association with patient outcome and chemotherapeutic response. Tripartite motif containing 28 (TRIM28) protein, a universal co-factor for KRAB-ZFP transcription factors [22], contributes to EMT and might be important for tumor metastasis in lung cancer [23]. Based on the link between KRAB-ZFPs and EMT [24], this study sought to explore whether NSCLC tumors with different EMT characteristics have a distinct survival outcome and the association of ZNF71 isoforms and EMT. Molecular functions of ZNF71 in NSCLC proliferation were evaluated with CRISPR-Cas9 and RNAi approaches. The overall study scheme is provided in Figure 1.

Specifically, mRNA expression of *ZNF71* isoforms and their prognostic implications were analyzed with public RNA-seq data of NSCLC patient tumors (*n* = 197) [25] and cell lines (*n* = 117) [26]. A 14-gene EMT classifier was constructed to evaluate the hybrid EMT states in NSCLC tumors, and *ZEB1* expression was used to evaluate the EMT states in cell lines. This EMT classification was further validated using stromal infiltration scores computed with software ESTIMATE [20]. The association between ZNF71 overall and isoform expression and EMT was examined in NSCLC tumors and cell lines. Functional assessment of ZNF71 in NSCLC proliferation was evaluated with public CRISPR-Cas9 [27] and RNAi [28] screening data. Finally, the association of the ZNF71 isoform and overall expression and chemoresponse to nine drugs commonly used to treat NSCLC was examined in cell lines (Figure 1).

## 2. Results

### 2.1. Expression of ZNF71 Isoforms in NSCLC Tumors and Cell Lines

ZNF71 gene is comprised of four exons, where exons 1 and 2 code for 180nt 5′UTR and the first 11 amino acids of the protein. Exon 3 codes for the next 43 amino acids (aa), encompassing most of the KRAB repression domain. Exon 4, the longest, codes for the remainder of the protein, including the 13-zinc-finger putative DNA binding domain and the predicted ~4 kb 3′UTR. The third KRAB-domain-containing exon could be alternatively spliced out to produce a KRAB-less isoform that will produce a shorter in-frame protein encoded by the last exon, i.e., exon 4. Approximately half of 800 human C2H2-type ZNF genes contain the evolutionary conserved one-exon-encoded KRAB domain, which could be alternatively spiced [9,10,11]. We analyzed ZNF71 isoform expression in the RNA-seq dataset GSE81089 of NSCLC tumor samples (*n* = 197) [25] and correlated the analysis with patient outcomes. Patient clinical characteristics is provided in Table A1.

The *ZNF71 KRAB* isoform (ZNF71_203_ENST00000599599) had a significantly higher expression (*p* < 0.001, *t*-tests) than the *KRAB*-less isoform (ZNF71_201_ENST00000328070) in NSCLC patient tumor samples (*n* = 197; Figure 2A) and cell lines (*n* = 117; Figure 2B). The expression of *ZNF71 KRAB* and *KRAB*-less isoforms was significantly correlated in both patient tumor samples (*p* < 8.6 × 10^−15^, Pearson’s correlation, Figure 2C) and cell lines (*p* < 2.3 × 10^−8^, Pearson’s correlation, Figure 2D). The expression of overall *ZNF71* and its isoforms was not significantly different among different histological subtypes in patient tumors (ANOVA tests; Figure A1). The expression of *ZNF71* overall and the *KRAB* isoform was significantly different among histological subtypes in NSCLC cell lines (*p* < 0.05, ANOVA tests; Figure A2). In the studied cell lines, large cell carcinoma had the highest *ZNF71* overall expression. Adenosquamous carcinoma had the highest *ZNF71 KRAB* expression. Squamous cell carcinoma had the lowest expression of both *ZNF71* overall and *KRAB* isoform (Figure A2).

To evaluate the prognostic performance of *ZNF71 KRAB*, patient tumor samples were divided into two groups using a cutoff of *ZNF71 KRAB* expression level measured with transcripts per million (TPM) of 1.5, which corresponds to the top 17% of *ZNF71 KRAB* expression versus the rest in the patient cohort. The cutoff value of TPM of 1.5 was chosen because TPM values are commonly normalized to the value of either 1 or 2 in RNA-seq raw data processing. The results showed that when *ZNF71 KRAB* isoform was expressed higher, patients survived for a significantly shorter time (log-rank *p* = 0.04, Kaplan–Meier analysis; Figure 2E), with a hazard ratio of 1.686 [1.026, 2.771]. In contrast, *ZNF71* overall expression or *ZNF71 KRAB*-less isoform expression did not generate significant prognostic stratification in the patient cohort using the same cutoff (top 17% expression level versus the rest) in Kaplan–Meier survival analyses (Figure 2F,G). These results indicate that the *ZNF71 KRAB* isoform is a more accurate prognostic factor for NSCLC than *ZNF71* overall expression and the *KRAB*-less isoform.

### 2.2. Association of ZNF71 KRAB Isoform with EMT

The above RNA-seq results were confirmed in qRT-PCR assays, which showed higher *ZNF71 KRAB* expression than the *KRAB*-less isoform in NSCLC cell lines (Figure 3A). Unfortunately, the ZNF71 antibody (Abcam, Cambridge, UK; ab87250) used in our previous study [6] was discontinued. We tried several other commercially available ZNF71 antibodies, but none of them was able to detect ZNF71 protein in our panel of NSCLC cell lines (not shown). After overexpression, ZNF71 protein expression (GeneTex, Irwin, CA, USA; Cat. No. GTX116553) was observed in HEK-293T cells (Figure 3B) in Western blots. The EMT properties of the NSCLC cell lines were tested using 3 mesenchymal markers (ZEB1, VIM, and FN1) and 11 epithelial markers (CDH1, EPCAM, ESRP1, ESRP2, DDR1, CTNNB1, CD24, CLDN7, KRT8, KRT19, and RAB25) using Western blots (Figure 3B). These 14 EMT markers were used to build an EMT classifier to divide patient samples into four groups based on the rank of the average expression values of all markers (Figure 3C). The average expression of 3 mesenchymal markers and that of 11 epithelial markers were computed for each patient sample, respectively. Based on the average epithelial and mesenchymal expression rank, patient samples were categorized into four EMT phenotypes: ranked as top 50% in both mesenchymal and epithelial (named High expression overlap), ranked as top 50% in mesenchymal but bottom 50% in epithelial (named Mesenchymal), ranked as top 50% in epithelial but bottom 50% in mesenchymal (named Epithelial), and ranked as the bottom 50% in both mesenchymal and epithelial (named Low expression overlap). A heatmap of the expression of EMT markers in categorized patient samples is shown in Figure 3D.

Patients defined by the four EMT phenotypes had significantly different disease-specific survival (log-rank *p*<0.01, Kaplan–Meier analysis). The patients in the High expression overlap group had the worst prognosis, with the shortest survival time, whereas the patients in the Epithelial group had the best prognosis, with the longest survival time (Figure 3E). The four EMT phenotypes were independent of patient cancer stage (*p* = 0.8569, chi-square tests; Table A2). The EMT phenotype was associated with *ZNF71 KRAB* expression (*p* = 0.0099, chi-square tests; Table A3). Within each EMT phenotype, we further performed a Kaplan–Meier survival analysis for the high *ZNF71 KRAB* isoform expression group (TPM ≥ 1.5) versus the low *ZNF71 KRAB* isoform expression group (TPM < 1.5) to assess the association of *ZNF71 KRAB* expression and patient survival in each EMT phenotype. In Epithelial and High expression overlap phenotypes, patients with higher KRAB expression had a significantly lower survival probability than those with lower KRAB expression (Figure 3H,I). In the Mesenchymal and Low expression overlap phenotypes, there was no significant difference in patient survival between the high and low KRAB expression groups (Figure 3F,G). In the Mesenchymal group, tumors with higher ZNF71 KRAB expression had a worse prognosis (no statistical significance; Figure 3F). In the Low expression overlap group, tumors with higher ZNF71 KRAB expression had a better prognosis (no statistical significance; Figure 3G). These results suggest that the *ZNF71 KRAB* isoform is a poor-prognosis marker for NSCLC tumors with high expression of epithelial markers but not for NSCLC with low expression of epithelial markers.

*ZNF71* overall expression and the *ZNF71 KRAB* and *KRAB*-less expression were all significantly different among all four EMT phenotypes in NSCLC tumors (*p* < 0.01, ANOVA tests; Figure 4A–C). Furthermore, Tukey’s honestly significant difference post-hoc test was performed among the EMT phenotypes. The expression of *ZNF71* (overall) and its isoforms was significantly lower in the Low expression overlap patient group than in the Epithelial phenotype, and the expression of *ZNF71* (overall) and the *KRAB* isoform was also significantly lower in the Low expression overlap patient group than in the High expression overlap patient group (*p* < 0.01, Tukey’s tests; Figure 4A–C). In Epithelial and High expression overlap phenotypes with higher expression of epithelial markers, the percentage of *ZNF71 KRAB* isoform high expression (TPM > 1.5) was greater than that in the other two EMT phenotypes (Figure 4D).

Solid tumors contain both epithelial and stromal cells, complicating the analysis of gene expression in patient tumor samples. On the contrary, NSCLC cell lines are devoid of any stromal components. Median *ZEB1* gene expression was used to categorize a panel of NSCLC cell lines (*n* = 117) [26] into two groups: *ZEB1* bottom 50% (more epithelial) and *ZEB1* top 50% (more mesenchymal) [19]. The NSCLC cell lines with higher ZEB1 expression (*ZEB1* top 50%) also had significantly higher expression of *ZNF71* overall, *KRAB*, and *KRAB*-less isoforms compared to the cell lines that had lower *ZEB1* expression (*p* < 0.05, two-sample *t*-tests; Figure 4E). These more mesenchymal cell lines would NOT be comparable to patient tumors with the Epithelial phenotype. A negative association of *ZNF71 KRAB* expression and patient survival was observed in NSCLC tumors with the Epithelial phenotype (Figure 3H).

The association between *ZNF71* and EMT in RNA-seq data was further analyzed with qRT-PCR and Western blots. The cell lines included in the qRT-PCR analysis (Figure 3A) can be categorized into four groups based on the protein expression of epithelial and mesenchymal markers in Western blots (Figure 3B): Epithelial (H441, H513, H820, and H358), Mesenchymal (H23, H460, and H1299), and Mostly Mesenchymal (A549 and H1395). *ZNF71 KRAB* and *KRAB*-less expression was significantly different among the three NSCLC cell line groups: Epithelial, Mesenchymal, and Mostly Mesenchymal (*p* < 0.05, ANOVA tests). *ZNF71 KRAB* expression was significantly higher in the Mesenchymal group than in the Epithelial group (*p* < 0.05, Tukey’s tests; Figure 4F). *ZNF71 KRAB*-less expression was significantly higher in the Mesenchymal group than in the Epithelial group and the Mostly Mesenchymal group (*p* < 0.05, Tukey’s tests; Figure 4F). When mesenchymal and mostly mesenchymal groups were combined in the analysis, only *ZNF71 KRAB* had a significantly higher expression in Mesenchymal/Mostly Mesenchymal group than in the Epithelial group (*p* < 0.05, two-sample *t*-tests; Figure 4G). Thus, the qRT-PCR and Western blot results further substantiated the association between *ZNF71 KRAB* and EMT observed in the RNA-seq data in NSCLC patient tumors and cell lines.

Stromal infiltration of the patient samples was assessed using the stromal scores computed with software ESTIMATE [20]. The patient tumors showing the highest average stromal score were in the Mesenchymal group, followed by High expression overlap, Low expression overlap, and Epithelial groups in descending order. The stromal scores of patient tumors defined as four EMT phenotypes were significantly different from each other (*p* < 0.001, ANOVA tests; *p* < 0.01, Tukey’s tests), except for the High-expression-overlap phenotype versus the Low-expression-overlap phenotype (Figure 4H). Since the NSCLC cell lines do not contain stromal cells, their average stromal scores are all negative (Figure 4I). Correlation of EMT markers and *ZNF71* isoforms with stromal and immune infiltration scores in patient tumors is provided in Table A4. These results validated our EMT classification of NSCLC tumors and cell lines.

### 2.3. Functional Analysis of ZNF71 in NSCLC Cell Lines

The functional role of ZNF71 in cell proliferation was evaluated in publicly available high-throughput CRISPR-Cas9 (*n* = 78) and RNAi (*n* = 92) screening data in NSCLC cell lines. Normalized ZNF71 dependency scores were not significant (below the threshold of –0.5) in the NSCLC cell lines in either RNAi (processed with DEMETER2) or CRISPR-Cas9 (Figure 5), indicating that a knockdown/knockout of ZNF71 did not significantly affect NSCLC tumor cell growth in vitro.

### 2.4. Association of ZNF71 Isoforms with Chemoresponse

Our previous study found that *ZNF71* overall gene expression was positively associated with prolonged survival in NSCLC patients who received cisplatin, carboplatin, and Taxol, indicating its association with chemosensitivity [6]. Drug responses of nine commonly used chemotherapeutic regimens in treating NSCLC were included in this study: carboplatin, cisplatin, paclitaxel (Taxol), docetaxel, gemcitabine, vinorelbine, etoposide, gefitinib, and erlotinib. The studied NSCLC cell lines include adenocarcinoma, squamous cell carcinoma, large cell carcinoma, and adenosquamous carcinoma (Figure A2). Due to the synergism and successful results of the combination of cisplatin–etoposide in treating small-cell lung cancer, long-term daily administration of oral etoposide in combination with cisplatin was used to treat NSCLC [29]. A systematic review showed that cisplatin–etoposide have efficacy comparable to that of carboplatin–paclitaxel when used with concurrent radiotherapy for patients with stage 3 unresectable NSCLC [30]. Gefitinib and erlotinib are widely used epidermal growth factor receptor (EGFR) tyrosine kinase inhibitors for treating advanced NSCLC with proven efficacy. A recent meta-analysis showed that gefitinib and erlotinib have comparable effects on patient survival, overall response rate, and disease control rate, with no considerable variation with regard to EGFR mutation status, ethnicity, line of treatment, and baseline brain metastasis status [31]. EGFR mutation in the studied NSCLC cell lines is provided in Table A5. EGFR mutation in sensitive and resistant cell lines in response to treatment with gefitinib and erlotinib, respectively, is provided in Table A6.

Differential expression of *ZNF71* and its isoforms was associated with chemoresponse to three drugs in the studied NSCLC cell lines. *ZNF71* overall and *KRAB* expression was significantly higher in docetaxel-resistant cell lines than docetaxel-sensitive cell lines assessed with the IC_50_ values (*p* < 0.05, two-sample *t*-tests; Figure 6). Docetaxel offers clinical benefits as a second-line treatment of NSCLC in patients previously treated with platinum-based chemotherapy [32]. It was recently reported that the combination of pembrolizumab (anti-PD1 immunotherapy) and docetaxel was well tolerated and substantially improved progression-free survival and overall response rate in patients with advanced NSCLC after platinum-based chemotherapy, including patients with EGFR variations [33]. *ZNF71 KRAB*-less was expressed higher in gemcitabine-sensitive cell lines than gemcitabine-resistant lines assessed with both IC_50_ and EC_50_ values (*p* < 0.05, two-sample *t*-tests; Figure 7 and Figure 8). Gemcitabine, a pyrimidine nucleoside antimetabolite, has been one of the most effective agents for treating advanced NSCLC [34]. *ZNF71 KRAB* was expressed significantly higher in paclitaxel-resistant cell lines than paclitaxel-sensitive lines assessed with the IC_50_ values (*p* < 0.05, two-sample *t*-tests; Figure 9). Paclitaxel, a tublin-binding agent, is commonly used to treat NSCLC in combination with a platinum-based compound [35].

H460 was categorized as having a partial response to paclitaxel in the studied NSCLC cell line panel (Figure 9A). Thus, the above-observed overexpression of *ZNF71 KRAB* in paclitaxel (Taxol)-resistant versus paclitaxel (Taxol)-sensitive cell lines did not include H460. Next, the expression of *ZNF71* and its isoforms was investigated in Taxol-resistant H460 cells (H460-R) [36] and parental cells (H460-P). *ZNF71* overall expression was lower in H460-R than H460-P cells (*p* < 0.05, two-sample *t*-tests; Figure 10). *ZNF71 KRAB* and *KRAB*-less expression had a similar differential expression pattern but was not statistically significant. These results were consistent with the observed positive correlation between *ZNF71* overall expression and prolonged survival in NSCLC patients who received cisplatin, carboplatin, and Taxol [6].

Overall, the observed overexpression of the *ZNF71 KRAB* isoform in tumor cells resistant to docetaxel and paclitaxel treatment compared to chemo-sensitive cells is consistent with its negative association with patient survival in NSCLC.

## 3. Discussion

Lung cancer is difficult to manage in clinics due to its complex etiology and somatic mutations. There are currently no clinically available gene tests to predict metastasis and clinical benefits of chemotherapy for all NSCLC, stages 1 to 3A. The 7-gene assay has been validated in our previous study as prognostic and predictive of chemotherapeutic benefits in multiple U.S. hospitals and a clinical trial JBR.10. (*n* = 331) [6]. The 7-gene assay also estimates each individual patient’s response to four drugs used to treat lung cancer: cisplatin, carboplatin, paclitaxel, and pemetrexed [6]. Thus, the 7-gene assay could meet the critical need in clinics to identify specific NSCLC patients who are at risk for tumor recurrence/metastasis and would benefit from receiving adjuvant chemotherapy. In addition, this capacity would reduce the use of adjuvant therapy in circumstances in which no benefit, but potentially negative side effects, would result.

Many studies have investigated molecular alterations in single genes in early-stage NSCLC and their prognostic and predictive implications [37], such as mutations in K*RAS*, *P53*, *EGFR* [38], *STK11* [39], mRNA, and protein expression of ERCC1 [40,41]. Nevertheless, none of these genes are ready for primetime clinical applications as prognostic and predictive biomarkers for early-stage NSCLC [37]. Several prognostic/predictive gene expression signatures for early-stage NSCLC have been reported [42,43,44,45], among which the Razor 14-gene assay [44,46] (Razor Genomics, Brisbane, CA, USA) has been commercialized for the prognosis of early-stage non-squamous NSCLC and is currently recruiting patients for a clinical trial. However, the Razor 14-gene assay is not applicable for squamous cell lung carcinoma, which accounts for 26% of lung cancer cases [47]. The Myriad myPlan™ Lung Cancer and Pervenio™ Lung RS tests were commercially available in clinics for prognosis of early-stage NSCLC [48]. However, they are no longer listed in the Myriad All Products page [49]. FoundationOne CDx (Foundation Medicine, Inc, Cambridge, MA, USA) and Oncomine DX (Thermo Fisher Scientific Inc., Waltham, MA, USA) receive reimbursement coverage to match stage 4 NSCLC patients to specific proteasome inhibitor-based therapy based on their tumor genetic mutations. However, they do not predict metastasis for patients in stages 1 to 3A.

Within the 7-gene signature, mRNA of *ZNF71* overall expression measured in qRT-PCR was positively correlated with survival in patients who received cisplatin, carboplatin, and Taxol, indicating that *ZNF71* mRNA is associated with chemosensitivity. Protein expression of ZNF71 was positively correlated with patient survival in independent tissue microarray cohort studies. However, mRNA of *ZNF71* overall expression measured in qRT-PCR was not found to be associated with NSCLC patient survival in the overall studied cohorts in our previous analysis [6]. Zinc finger proteins (ZFPs) are the largest family of transcription factors not only in human cells but also in most eukaryotes, representing about 3% of the total human genome [7,50]. A repeating motif within ZFPs contains two histidine and two cysteine amino acid residues (i.e., C2H2) coordinating zinc ion and has the ability to bind to DNA, RNA, or cellular proteins [8]. Therefore, ZFPs have a wide range of predicted functions according to their molecular structure, including DNA repair, degradation of proteins, signal transductions, migration of cells, regulation of apoptosis, lipid binding, and transcription regulation [7,8]. KRAB-ZFPs are a family of transcriptional repressors with diverse functions, most notably the silencing of transposable elements [51]. They contain an N-terminal KRAB domain and a C-terminal C2H2-type zinc finger array. The repressor action of KRAB-ZFPs requires the recruitment of KRAB-associated protein 1, also known as tripartite motif protein 28, KAP1/TRIM28. KAP1 functions as a scaffold complex composed of histone methyl transferase (SETDB1), heterochromatin protein-1 (HP-1), nucleosome remodeling and deacetylation (NuRD), and DNA methyl transferase [11]. When KRAB-ZFPs recruit KAP-1, the formed repressor complex leads to heterochromatin formation. Our group reported that KAP1 promotes proliferation and metastatic progression of breast cancer cells [52], consistent with the observed association between KAP1 and breast cancer progression [53]. RB-associated KRAB zinc finger (BRAK) was upregulated in NSCLC and was associated with poor prognosis in patients [52]. Zinc finger protein 668 (ZNF668) was reported to suppress NSCLC invasion and migration by down-regulating Snail and upregulating E-cadherin and zonula occludens-1 [54]. Down-regulated protein expression of ZNF668 was found in NSCLC tumors compared with normal lung tissues, and there was a negative association between ZNF668 protein expression and lymph node metastasis [54].

In this study of public RNA-seq data, *ZNF71 KRAB* had a significantly higher expression than the *ZNF71 KRAB*-less isoform in NSCLC patient tumors [25] and cell lines [27]. The expression of both isoforms was significantly correlated in patient tumors and cell lines. Patients with higher *ZNF71 KRAB* expression had a significantly worse survival outcome than patients with lower *ZNF71 KRAB* expression, whereas the *ZNF71* overall expression and the *ZNF71 KRAB*-less isoform were not prognostic in the same patient cohort (Figure 2). In this study, *ZNF71* overall expression was not prognostic in RNA-seq data of NSCLC patient cohorts, which is consistent with our previous qRT-PCR results. We designed primers for *ZNF71 KRAB* and *KRAB*-less isoforms for TaqMan qRT-PCR assays, and the results in NSCLC cell lines confirmed the overall higher expression of *ZNF71 KRAB* than that of the *KRAB*-less isoform. Further investigation revealed an association between *ZNF71 KRAB* expression and EMT in RNA-seq data of NSCLC patient tumors and cell lines, which was validated in qRT-PCR and Western blot assays of cell lines (Figure 3 and Figure 4). *ZNF71 KRAB* was overexpressed in cell lines resistant to docetaxel (Figure 6) and paclitaxel (Figure 9) treatment compared to chemo-sensitive cell lines, which was consistent with its association with poor prognosis in patients. Given the transcriptional repression role of the KRAB domain, the negative correlation between *ZNF71 KRAB* expression and NSCLC patient survival could be reasonable. These results indicate that the ZNF71 KRAB isoform is a more effective prognostic factor than *ZNF71* overall expression and the *ZNF71 KRAB*-less isoform for NSCLC. Further investigation is warranted to explore whether the *ZNF71 KRAB* isoform provides added prognostic and predictive value to the original 7-gene assay for NSCLC or whether *ZNF71 KRAB* could replace *ZNF71* overall expression in the 7-gene assay to achieve better prognostic performance.

The expression of overall *ZNF71* and its isoforms was not significantly different among different histological subtypes in patient tumors (Figure A1). The expression of *ZNF71* overall and the *KRAB* isoform was significantly different among histological subtypes in NSCLC cell lines (Figure A2). The observed discrepancy in *ZNF71* overall and *KRAB* expression among NSCLC histological subtypes in patient tumors and cell lines is possibly due to the fact that stromal cells are present in the NSCLC tumors but not in the studied cell lines. There was a strong association between *ZNF71* overall and *KRAB* expression and stromal infiltration in patient tumors (Table A4) but not in NSCLC cell lines (results not shown). In the future, it would be interesting to investigate whether *ZNF71* overall and *KRAB* expression could differentiate NSCLC histological subtypes in micro-dissected epithelial cells from NSCLC tumors.

The role of EMT in cancer patient outcomes is not well defined and remains controversial [19,55]. In this study, we developed an EMT classifier based on transcriptional profiles of 14 EMT markers, and NSCLC patients defined with four EMT phenotypes had distinct survival outcomes (Figure 3). Out of the 14 EMT markers, 13 (all except for *CTNNB1*) used in our classification were included in the pan-cancer EMT classifier presented in Panchy et al. [21], which also divides tumors into quadrants to characterize hybrid EMT states in tumors. Our EMT classifier was validated using stromal scores quantified with the well-established software ESTIMATE [20]. *ZNF71 KRAB* expression was prognostic in NSCLC patients with high expression of EMT markers (the Epithelial group and the High expression overlap group) but not prognostic in tumors with low expression of EMT markers (the Mesenchymal group and the Low expression overlap group). These results imply that transcriptional biomarkers might have distinct prognostic implications in tumors with different EMT characteristics. The unclear role of EMT in patient outcomes is further complicated by tumor stromal and immune infiltration. For instance, the expression of *ZNF71* overall and its isoforms was not significantly different in NSCLC patient tumors separated by median *ZEB1* expression (the top 50% *ZEB1* expression group versus the bottom 50% *ZEB1* expression group, two-sample *t*-tests; results not shown). However, a strong association was found between *ZNF71 KRAB* and *ZEB1* gene expression in NSCLC epithelial cell lines (Figure 4E).

ZNF71 molecular functions have not been reported in the literature. This study evaluated the functional associations of ZNF71 using publicly available genome-scale CRISPR-Cas9 and RNAi screening data. The knockouts and knockdowns in NSCLC cell lines were performed for overall ZNF71, i.e., all transcripts. They did not affect cell proliferation (Figure 5). Concordantly, overall *ZNF71* mRNA expression was not associated with patient survival. We found that *ZNF71 KRAB* was associated with patient outcome and with EMT, i.e., expressed higher in the top 50 *ZEB1* expressing cells. A mechanistic link between ZNF71 KRAB and EMT is currently not known. We could speculate that EMT is associated with significant changes in splicing and hence can potentially skew splicing for inclusion of the KRAB domain [56,57]. The link between other KRAB-ZFPs and EMT was reported. TRIM28 protein, a universal co-factor for KRAB-ZFP transcription factors [22], is known to participate in a wide range of aspects of cellular biology, either promoting cell proliferation [52] or mediating anti-proliferative activities [58]. TRIM28 protein is involved in cancer by regulating gene expression through heterochromatin formation, mediation of DNA damage response, inhibition of p53 activity, regulation of EMT, and maintenance of stem cell pluripotency and genome stability [59]. TRIM28 expression is induced following transforming growth factor-β (TGF-β) treatment at both protein and mRNA levels. TRIM28 deficiency impairs TGF-β-induced EMT and decreases cell migration and invasion, and the expression of TRIM28 affects the acetylation of histones on E-cadherin and N-cadherin promoters, suggesting that TRIM28 contributes to EMT and might be important for tumor metastasis in lung cancer [23]. ZNF382 KRAB regulates EMT and functions as a tumor suppressor in gastric cancer [24]. To gain better insight into the mechanistic link between ZNF71 KRAB and EMT, we are currently conducting genome-scale network analysis to identify all the genes showing a significant statistical association with *ZNF71* at gene expression and DNA copy number variation levels in NSCLC patient tumors. We have identified ZNF71-mediated molecular association networks in EMT using public data. A similar analysis will be conducted to identify ZNF71 KRAB mediated molecular networks in EMT using public RNA-seq data generated from NSCLC patient tumors. In the future knockdown/overexpression experiments, we will examine which of these EMT-relevant genes in the identified networks are affected by ZNF71 KRAB to investigate the potential mechanistic link between ZNF71 KRAB and EMT in NSCLC.

The association between ZNF71 and its isoforms in terms of chemoresponse is also complex due to different genetic predisposition to chemotherapeutic regimens in tumor cells. In the studied NSCLC cell line panel, *ZNF71 KRAB* overexpression was observed in tumor cells that were resistant to Taxol compared to Taxol-sensitive tumor cells (Figure 9). These results were consistent with the observed negative association of *ZNF71 KRAB* expression and patient survival (Figure 3). In the H460 cell line that is defined as having a partial response to Taxol in the studied cell line panel (Figure 9A), *ZNF71* overall expression was significantly lower in H460-R than in H460-P cells (Figure 10). These results were consistent with the observed association of *ZNF71* overall expression with chemosensitivity to Taxol in NSCLC patients [6]. Significantly higher *ZNF71* overall and *KRAB* expression in docetaxel-resistant cell lines suggests their potential use in predicting predisposition to this agent or in combined immunotherapy and chemotherapy, based on the recent report of the efficacy of the combination of docetaxel and pembrolizumab in treating NSCLC [33].

*ZNF71* overall and *KRAB* expression was significantly higher in a normal lung small airway epithelial cell line (SAEC) compared with that in most NSCLC cell lines analyzed in qRT-PCR. Since we cannot draw any solid conclusion based on one normal lung cell line, we did not show these results. One possible explanation could be that SAEC cells were cultured in specialized media (Lonza, Basel, Switzerland) with added defined growth factors, while the lung cancer cell lines were cultured in standard DMEM plus 10% FBS. Many KRAB-ZFPs were reported to act as either tumor suppressors or oncogenes [60]. ZNF382 is down-regulated in multiple carcinoma types due to promoter methylation and functions as a tumor suppressor in gastric cancer [24]. ZNF23, a KRAB-containing protein, is down-regulated in human cancers and inhibits cell cycle progression [61]. RB-associated KRAB (RBAK) zinc finger is upregulated in NSCLC and promotes cell migration and invasion [62]. We are planning to carry out the following analysis of ZNF71 overall and isoform expression in our future study: (1) examine multiple normal lung epithelial cell lines in qRT-PCR, (2) analyze TCGA data for NSCLC tumors versus normal lung tissue samples, and (3) design knockdown and overexpression of ZNF71 in vitro and/or in vivo xenograft studies to examine whether it is oncogenic or tumor suppressive.

## 4. Materials and Methods

### 4.1. NSCLC Patient Samples and RNA-Seq Data

A total of 199 NSCLC patient tumors were collected in a previous study [25]. Patient clinical information is provided in Table A1. Deep RNA sequencing of the patient tumors was generated with Illumina HiSeq 2500 (raw data available at NCBI GEO with accession number GSE81089). Patients with sufficient survival information (*n* = 197) were included in this study. The raw RNA-seq data were processed with Salmon to quantify the expression of *ZNF71* at the isoform level by using transcripts per million (TPM) reads [63].

### 4.2. Cancer Cell Line Encyclopedia (CCLE)

Gene expression data for CCLE were downloaded from DepMap 20Q2 (https://figshare.com/articles/dataset/DepMap_20Q2_Public/12280541, accessed on 1 April 2021) [26]. Gene expression data were obtained from the CCLE data portal (https://data.broadinstitute.org/ccle/CCLE_RNAseq_081117.rpkm.gct, accessed on 1 April 2021). RNA-seq data were quantified using the GTEx pipelines [64]. A total of 117 NSCLC cell lines were included in this analysis.

### 4.3. CRISPR-Cas9 Assays

Gene knockout effects in CCLE using CRISPR-Cas9 screens were quantified in Project Achilles [27,65]. The data were obtained from DepMap 20Q2 (https://figshare.com/articles/dataset/DepMap_20Q2_Public/12280541, accessed on 1 April 2021) [26]. The CRISPR-Cas9 data were processed with the CERES method [27]. Gene effects in each cell line were normalized such that the median non-essential gene knockout effect is 0 and the median essential gene knockout effect is -1. A gene is defined as an essential gene if it is essential to the cell growth in each line; otherwise, it is defined as a non-essential gene. There were 78 NSCLC cell lines with genome-scale CRISPR-Cas9 knockout results.

### 4.4. RNAi Functional Assays

Genome-scale RNAi screening data in CCLE were obtained from Project Achilles [28] (https://depmap.org/R2-D2/, accessed on 1 April 2021). The DEMETER2 method [28] was used to estimate average gene dependency scores in each cell line for short hairpin RNA (shRNA) libraries. Gene dependency scores were standardized with DEMETER2 such that the median of the across-cell-line average dependency scores of the positive control gene set was -1 and that of the negative control gene set was 0. There were 92 NSCLC cell lines with genome-scale RNAi screening results normalized with DEMTER2.

### 4.5. Drug Response

The growth inhibitory activity of 4518 drugs was quantified in 578 human cancer cell lines using the PRISM molecular barcoding and multiplexed screening method [66]. The PRISM repurposing dataset is available at the Cancer Dependency Map portal (https://depmap.org/portal/download/, accessed on 1 April 2021). Drug responses of nine commonly used chemotherapeutic regimens in treating NSCLC were included in this study: carboplatin, cisplatin, paclitaxel, docetaxel, gemcitabine, vinorelbine, etoposide, gefitinib, and erlotinib. For each drug, cell lines were defined as resistant, sensitive, or partial response by using the mean ± 0.5 standard deviation (SD) of the IC_50_ or EC_50_ values [67,68]. Cell lines with an IC_50_ or EC_50_ value greater than the mean + 0.5 SD were defined as resistant to the drug. Cell lines with an IC_50_ or EC_50_ value less than the mean − 0.5 SD were defined as sensitive to the drug, and those with an IC_50_ or EC_50_ value between the mean + 0.5 SD and the mean − 0.5 SD were defined as having a partial response to the drug. This categorization corresponds to the RECIST 1.1 system (i.e., complete response, partial response, and stable disease/disease progression) in evaluating chemotherapeutic response in solid tumors [69].

### 4.6. RNA Extraction, Quality Assessment, and qRT-PCR

Total RNA was extracted from multiple cell lines using the RNeasy Mini kit (Qiagen, Hilden, Germany). According to manufactures’ protocol, RNA was eluted in 30 µL of deionized water and kept at −80 °C until use. The concentration of RNA was determined with NanoDrop, and RNA purity was verified by determining the A260/A230 ratio. One microgram of total RNA was reverse-transcribed using SuperScriptIII First Strand Synthesis SuperMix for qRT-PCR (Invitrogen, Carlsbad, CA, USA). For quantitative real-time PCR, either SYPR Select Master Mix (Thermo Fisher) or a TaqMan master mix (IDT) was used, depending on the primers/probes used.

All primers were purchased from IDT. There were three sets of ZNF71 primers: Set 1 (will amplify both isoforms of the gene) primer F: 5′- CAGCACTTCAGACCTCAGTAAG-3′, primer R: 5′-TTGGTGCTTTATCAGGGACG-3′; Set 2, KRAB-less ZNF71, primer F: 5′-GCCTGTCTTCCTATTCACCG-3′, primer R: 5′-CATTTCAGGTCTAGTCTCCCAG-3′, probe: /56-FAM/AGC CAT CCC /ZEN/TCT GCT GCC C/3IABkFQ/; Set 3, KRAB-ZNF71, primer F: 5′-GACGTTCAGGGATGTGACTG-3′, primer R: 5′TTCAGGTCTAGTCTCCCAGTC-3′, probe: /56-FAM/AGG TCC TTC /ZEN/TGG GCA GGC TC/3IABkFQ/.

The PCR reactions were loaded in a 384-well plate in triplicate. The expression level of *ZNF71* was calculated using the comparative threshold (2^^−∆∆Ct^) method and normalized to *UBC* or *RPL4* housekeeping genes: *UBC* primer set, primer F: 5′-GATTTGGGTCGCAGTTCTTC-3′, primer R: 5′-CCTTATCTTGGATCTTTGCCTTG-3′ and *RPL4* (Cat. N PPH13915A, SABiosciences, Federick, MD, USA). For SYBR Green, PCR melting curve analysis was performed to verify that the reactions had a single product. *UBC* was used as the housekeeping gene to calculate the delta Ct value for each sample in this study.

### 4.7. Western Blots

Cells were lysed in gel lysis buffer (GLB). GLB was composed of 50 mM Tris-HCl, pH 6.8, 2% SDS, and 10% glycerol. Total protein was quantified using the BCA assay (Pierce). An equal amount of protein lysate (20–30 µg) was loaded and separated on 4-12% Bis-Tris gels, then transferred to polyvinylidene difluoride (PVDF) membrane (Fisher), and probed with antibodies as previously described [70]. ZNF71 antibody was purchased from GeneTex (Cat. No. GTX116553). The following antibodies were included in Western blots: ZEB1 (Sigma, St. Louis, MI, USA; HPA027524), E-cadherin (BD Biosciences, 610181), ESRP1/2 (23A7) (Novus, Centennial, CO, USA; NBP1-77971), beta-Catenin (6B3, Cell Signaling, Danvers, MA, USA; 9582), DDR1 (D1G6, Cell Signaling, 5583), pan-Cytokeratin (C11, Santa Cruz, Dallas, TX, USA; sc-8018), Actin (I-19, Santa Cruz, sc-1616), PCNA (eBioSciences, 14-6748-81), GAPDH (Millipore, Burlington, MA, USA; MAB374), and secondary horse radish peroxidase (HRP)-conjugated antibodies against mouse and rabbit (Jackson ImmunoResearch, West Grove, PA, USA). Standard chemiluminescence was used to detect the protein bands.

### 4.8. Responses to Taxol with Annexin V Staining for Apoptosis Using Flow Cytometry

H460 cells (denoted as H460-P) and a derivative cell line H460-R, reported to be Taxol resistant [36], were plated in 6-well plates in three biological replicates and incubated with either 10 nM or 50 nM Taxol for 48 h, with DMSO used as vehicle control. Supernatants were centrifuged to collect floating cells which were combined with trypsinized cells and subsequently stained with Annexin V and PI. The percentage of apoptotic/dead cells was determined by flow cytometry. An Annexin V binding assay was performed using the FITC Annexin V/Dead Cell Apoptosis Kit with FITC Annexin V and PI for flow cytometry (Invitrogen). PI is the standard reagent to exclude non-viable cells from flow cytometry assay, i.e., to calculate cell death. Flow cytometry was performed at the Flow Cytometry and Single Cell Core Facility at West Virginia University on the LSRFortessa machine.

### 4.9. Statistical Analysis

Statistical analysis was performed using Rstudio version 1.1.456 [71]. Differential gene expression between two groups was evaluated with Student’s *t*-tests, and a two-sided *p*-value < 0.05 was considered statistically significant. Differences in expression among more than two groups were evaluated with ANOVA and Tukey’s honestly significant difference tests, and a *p*-value < 0.05 was considered as statistically significant. Survival analysis was performed using Kaplan–Meier analysis with the *survival* package in R. Log-rank tests were used to assess the difference in survival probability from different groups in Kaplan–Meier analyses. A heatmap was generated with the *heatmap.2* function from the *gplots* package in R. Partial correlation was used to find the relationship between two variables and eliminate the variance of the third variable using functions *pcor* and *pcor.test* in the *ppcor* package in R. A mixed-effect model was used to assess the difference between two conditions, using R package *Ime4*. A linear mixed-effect model was used to assess the difference between different conditions as fixed effects (biological samples as random effects), considering the correlation between technical replicates. The *p*-value was calculated based on asymptotic Z-distribution.

### 4.10. Assessment of Stromal Infiltration and Immune Infiltration

A gene’s stromal and immune infiltration in NSCLC tumors was evaluated with the Estimation of STromal and Immune cells in MAlignant Tumours (ESTIMATE) method [20]. The function *estimateScore* was performed on gene expression data to evaluate the stromal scores and immune scores in each sample with the *estimate* package in R.

### 4.11. ZNF71 Overexpression

For overexpression purposes, the ZNF71 Plasmid construct was purchased from the PlasmID database (https://plasmid.med.harvard.edu/PLASMID/OrderOverview.jsp, accessed on 1 April 2021) maintained by Harvard University (catalog number HsCD00412101). ZNF175 plasmid construct was used as a control (catalog number HsCD00421069), and pLU-GFP vector was also used as a control. For HEK-293T cell transfection, calcium phosphate was used as a transfection reagent, which is commonly used to introduce DNA into eukaryotic cells to obtain both transient and permanent transfections. The approach is based on mixing HEPES-buffered saline containing Na_3_PO_4_ with CaCl_2_ containing the DNA. The DNA calcium phosphate stick to the cell membrane then is taken by cellular endocytosis_._ For the NSCLC transfection reagent, Lipofectamine was purchased from Thermo Fisher Scientific (catalog number L3000008). Cells were cultured in 6-well plates seeding 400,000–500,000 cells per plate according to the transfection type: reverse, serum-free medium, or regular transfection. A total of 15 µg of plasmid in 50 µL of CaCl was used to transfect HEK-293 in a 35 × 10 mm cell culture dish.

## 5. Conclusions

This study revealed that the ZNF71 KRAB isoform is a more effective prognostic factor for NSCLC than ZNF71 overall expression and the ZNF71 KRAB-less isoform. ZNF71 KRAB was overexpressed in NSCLC cell lines resistant to docetaxel and paclitaxel treatment compared to chemo-sensitive cell lines, consistent with its association with a poor prognosis in patients. These results suggested that the ZNF71 KRAB isoform may provide added prognostic and predictive value to the 7-gene assay developed in our previous study [6] for NSCLC, upon further evaluation using patient samples. ZNF71 KRAB expression was also associated with EMT in both NSCLC patient tumors and cell lines. The knockdown/knockout of ZNF71 did not significantly affect cell growth in NSCLC cell lines, implying that ZNF71 might not be involved in cell proliferation. The results from this study provided evidence of potential implications of the ZNF71 KRAB isoform in NSCLC prognosis and enlightened its possible mechanism in EMT for future research.

## 6. Patents

The 7-gene NSCLC prognostic and predictive assay is included in U.S. National Phase Patent Application No. 17/251,359 and International Non-Provisional Patent Application No. PCT/US20/23597.

## Figures and Tables

**Figure 1 ijms-22-03752-f001:**
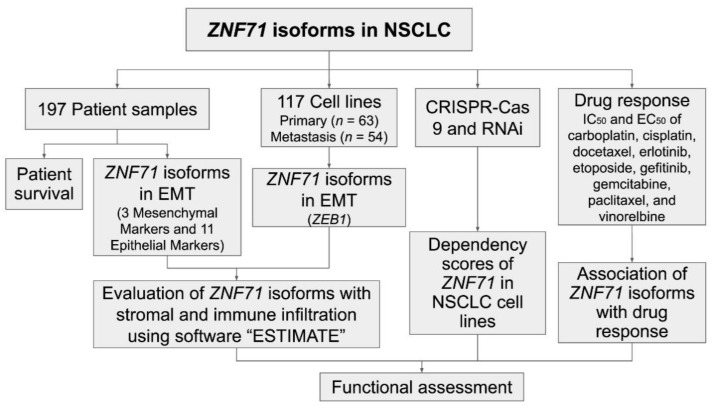
Overall study scheme. The arrows indicate analysis flow.

**Figure 2 ijms-22-03752-f002:**
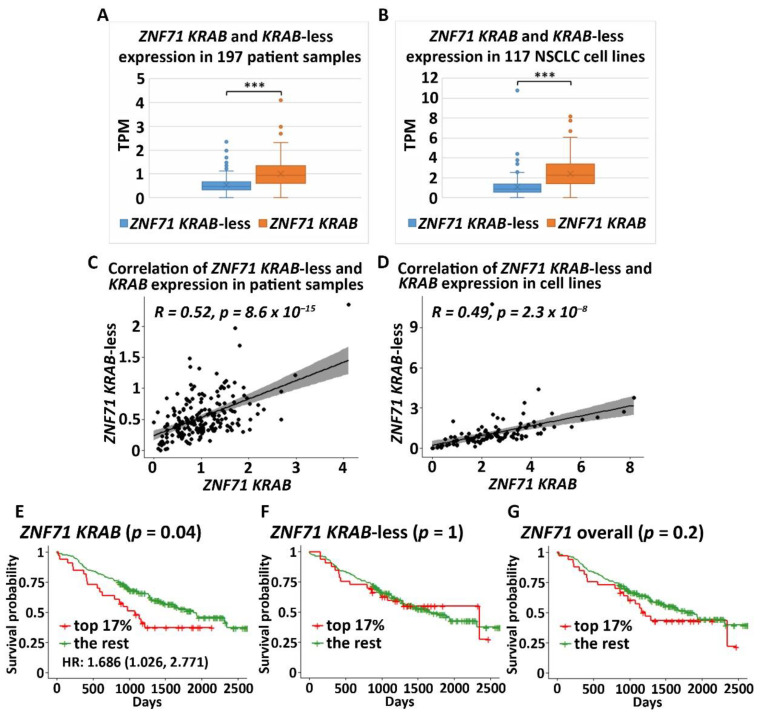
Zinc finger protein 71 Krüppel associated box (*ZNF71 KRAB*) and *KRAB*-less expression non-small-cell lung cancer (NSCLC) patient tumors and cell lines (**A**). Comparison of the expression of *ZNF71 KRAB* and *KRAB*-less isoforms in NSCLC tumors (*** *p* < 0.001, two-sample *t*-tests). (**B**) Comparison of the expression of *ZNF71 KRAB* and *KRAB*-less isoforms in NSCLC cell lines (*** *p* < 0.001, two-sample *t*-tests). (**C**) Pearson’s correlation of the expression of *ZNF71 KRAB* and *KRAB*-less isoforms in patient tumors. (**D**) Pearson’s correlation of the expression of *ZNF71 KRAB* and *KRAB*-less isoforms in cell lines. (**E–G**) Kaplan–Meier survival analyses of patient tumors grouped by *ZNF71 KRAB* isoform expression (**E**), *ZNF71 KRAB*-less isoform expression (**F**), and *ZNF71* overall expression (**G**), with top 17% expression versus the rest, respectively. When *ZNF71 KRAB* was expressed higher than transcripts per million (TPM) of 1.5 (top 17%), the patients survived for a significantly shorter length of time.

**Figure 3 ijms-22-03752-f003:**
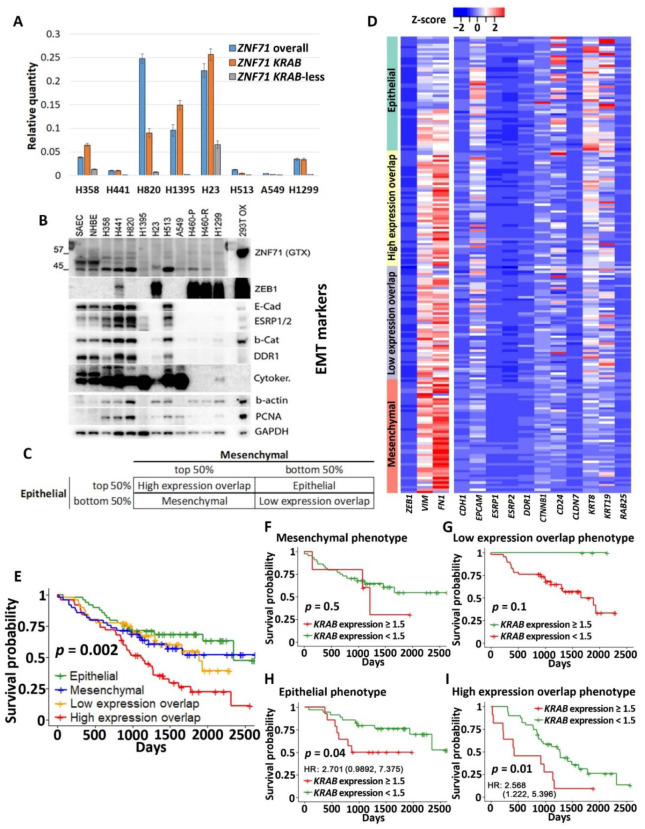
ZNF71 isoforms in epithelial-to-mesenchymal transition (EMT) and patient survival. (**A**) Relative quantity of *ZNF71*, *KRAB*, and *KRAB*-less in qRT-PCR assays of NSCLC cell lines. (**B**) Western blots of EMT markers as well as endogenous ZNF71 and overexpressed ZNF71 in HEK-293T (top lanes). (**C**) EMT classification of NSCLC tumors based on the average expression rank of epithelial and mesenchymal markers. The table shows the final four phenotype categorization of patient samples. (**D**) The expression of 14 EMT markers in patient tumors grouped by four phenotypes defined in D. (**E–I**) Kaplan–Meier survival analyses of patient tumors grouped by four EMT phenotypes (**E**) and in each phenotype (**F–I**). The Epithelial phenotype had the best patient survival outcome, and the High expression overlap phenotype had the worst patient survival outcome; *ZNF71 KRAB* is a poor prognosis marker in both phenotypes.

**Figure 4 ijms-22-03752-f004:**
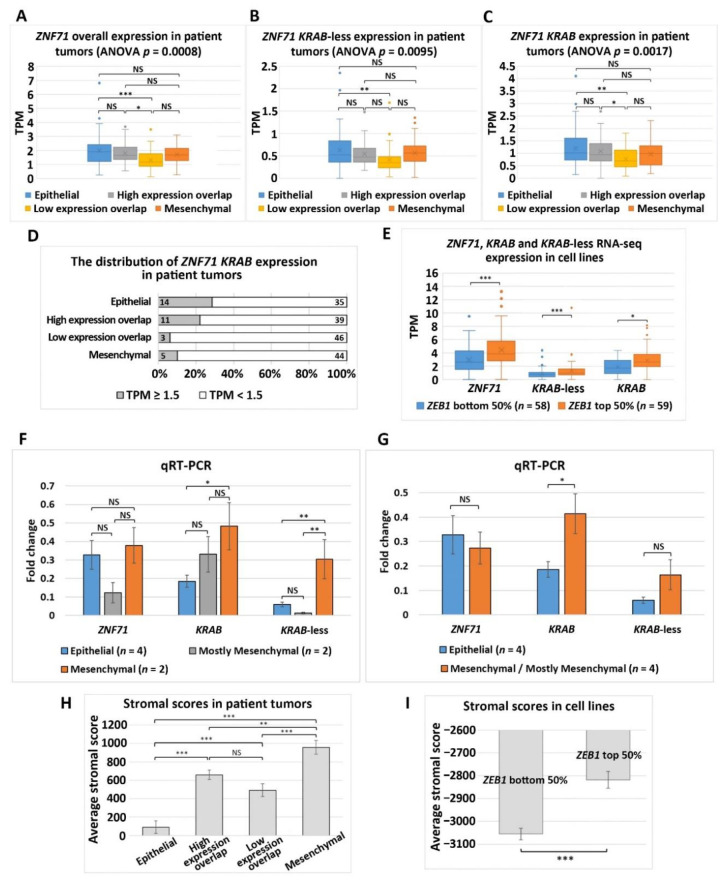
Association of *ZNF71* isoforms and EMT in NSCLC tumors and cell lines. (**A**–**C**) Comparison of *ZNF71* overall (**A**), *KRAB*-less isoform (**B**), and *KRAB* isoform (**C**) expression in patient tumors categorized by four EMT phenotypes. (**D**) Distribution of *ZNF71 KRAB* expression (TPM ≥ 1.5) versus *ZNF71 KRAB* expression (TPM < 1.5) in patient tumors defined by four EMT phenotypes. The number of patient samples in each category is provided in the figure. (**E**) Comparison of *ZNF71* overall, *KRAB*-less, and *KRAB* expression in NSCLC cell lines categorized by median ZEB1 expression. (**F**) Fold change of *ZNF71* overall, *KRAB*, and *KRAB*-less expression in Epithelial, Mesenchymal, and Mostly Mesenchymal NSCLC cell lines in qRT-PCR analysis (individual cell line results displayed in Figure 3A). * *p*-Value < 0.05, ** *p*-value < 0.01, and *** *p*-value < 0.001; NS (non-significance): *p*-value > 0.05 in ANOVA and Tukey’s tests. (**G**) Fold change of *ZNF71* overall, *KRAB*, and *KRAB*-less expression in Epithelial and Mesenchymal/Mostly Mesenchymal (combined) NSCLC cell lines in qRT-PCR analysis. * *p*-Value < 0.05, ** *p*-value < 0.01, and *** *p*-value < 0.001; NS: *p*-value> 0.05 in two-sample *t*-tests. (**H**,**I**) EMT classifiers in both patient tumors and cell lines were validated by stromal scores quantified with ESTIMATE [20]. (**H**) Comparison of stromal scores of patient tumors categorized by four EMT phenotypes. (**I**) Comparison of stromal scores of NSCLC cell lines categorized by median *ZEB1* expression.

**Figure 5 ijms-22-03752-f005:**
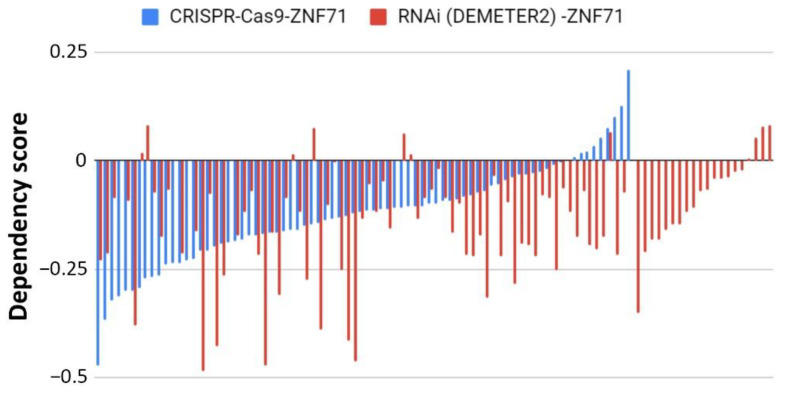
Functional analyses of ZNF71 in NSCLC cell lines using CRISPR-Cas9 and RNAi approaches. Dependency scores of ZNF71 in NSCLC cell lines using CRISPR-Cas9 (*n* = 78) and RNAi (normalized with DEMETER2; *n* = 92).

**Figure 6 ijms-22-03752-f006:**
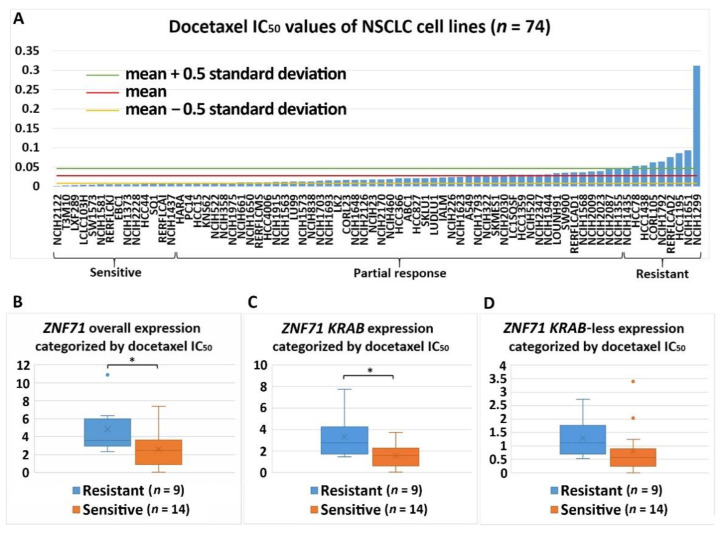
Association of the expression of *ZNF71* and its isoform with docetaxel drug response (IC_50_) in NSCLC cell lines (*n* = 74). *ZN**F71* overall and *KRAB* expression was significantly higher in docetaxel-resistant cell lines compared with docetaxel-sensitive cell lines. (**A**) Docetaxel IC_50_ values of NSCLC cell lines. (**B**–**D**) Comparison of *ZNF71* overall (**B**), KRAB-less isoform (**C**), and *KRAB* isoform (**D**) expression in cell lines categorized by IC_50_ values (resistant versus sensitive). * *p* < 0.05 in two-sample *t*-tests used in (**B**–**D**).

**Figure 7 ijms-22-03752-f007:**
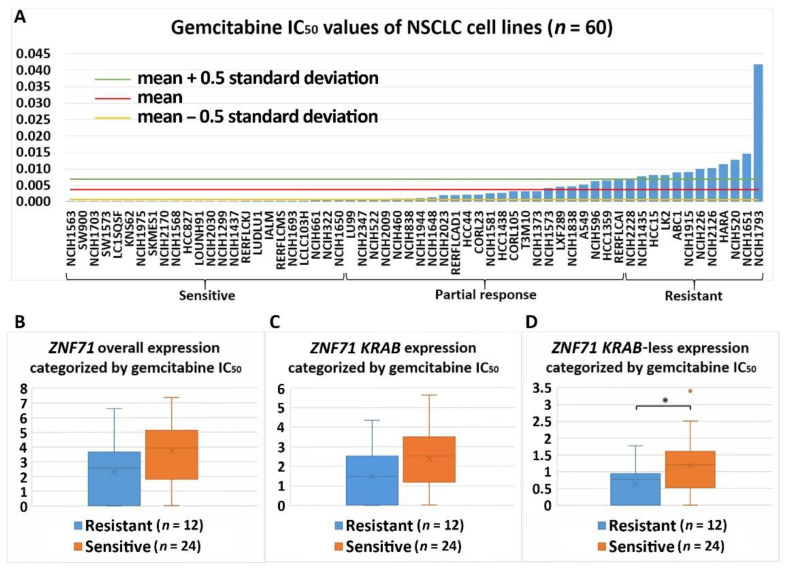
Association of the expression of *ZNF71* and its isoform with gemcitabine drug response (IC_50_) in NSCLC cell lines (*n* = 60). *ZN**F71 KRAB*-less expression was significantly higher in gemcitabine-sensitive cell lines compared with gemcitabine-resistant cell lines. (**A**) Gemcitabine IC_50_ values of NSCLC cell lines. (**B**–**D**) Comparison of *ZNF71* overall (**B**), *KRAB*-less isoform (**C**), and *KRAB* isoform (**D**) expression in cell lines categorized by IC_50_ values (resistant versus sensitive). * *p* < 0.05 in two-sample *t*-tests used in (**B**–**D**).

**Figure 8 ijms-22-03752-f008:**
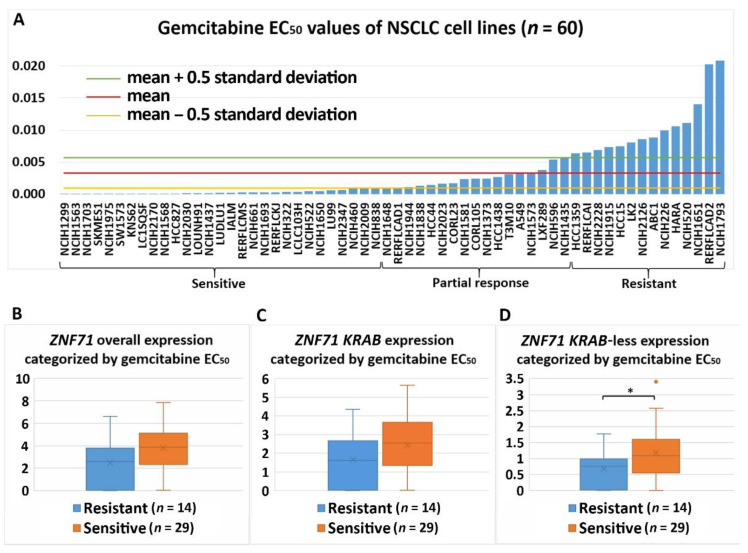
Association of the expression *ZNF71* and its isoform with gemcitabine drug response (EC_50_) in NSCLC cell lines (*n* = 60). *ZN**F71 KRAB*-less expression was significantly higher in gemcitabine-sensitive cell lines compared with gemcitabine-resistant cell lines. (**A**) Gemcitabine EC_50_ values of NSCLC cell lines. (**B**–**D**) Comparison of *ZNF71* overall (**B**), *KRAB*-less isoform (**C**), and *KRAB* isoform (**D**) expression in cell lines categorized by EC_50_ values (resistant versus sensitive). * *p* < 0.05 in two-sample *t*-tests used in (**B**–**D**).

**Figure 9 ijms-22-03752-f009:**
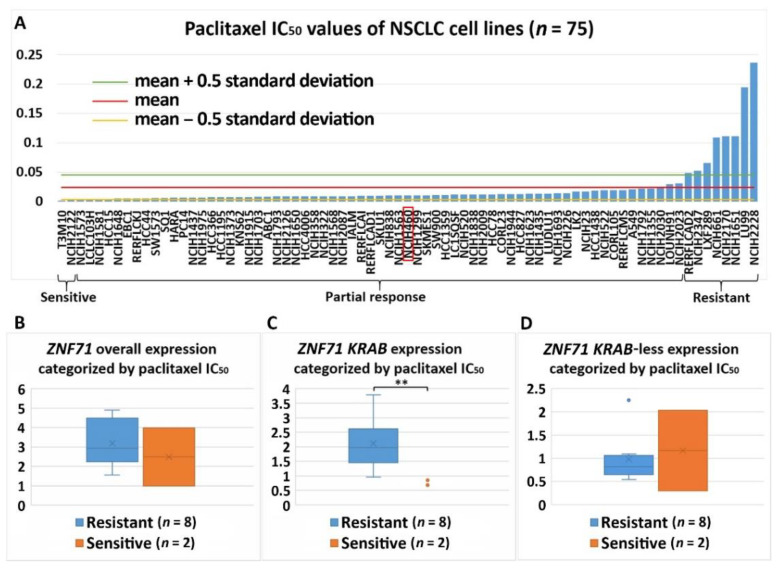
Association of the expression *ZNF71* and its isoform with paclitaxel drug response (IC_50_) in NSCLC cell lines (*n* = 75). *ZNF71 KRAB* expression was significantly higher in paclitaxel-resistant cell lines compared with paclitaxel-sensitive cell lines. (**A**) Paclitaxel IC_50_ values of NSCLC cell lines. (**B**–**D**) Comparison of *ZNF71* overall (**B**), *KRAB*-less isoform (**C**), and *KRAB* isoform (**D**) expression in cell lines categorized by IC_50_ values (resistant versus sensitive). ** *p* < 0.01 in two-sample *t*-tests used in (**B**–**D**).

**Figure 10 ijms-22-03752-f010:**
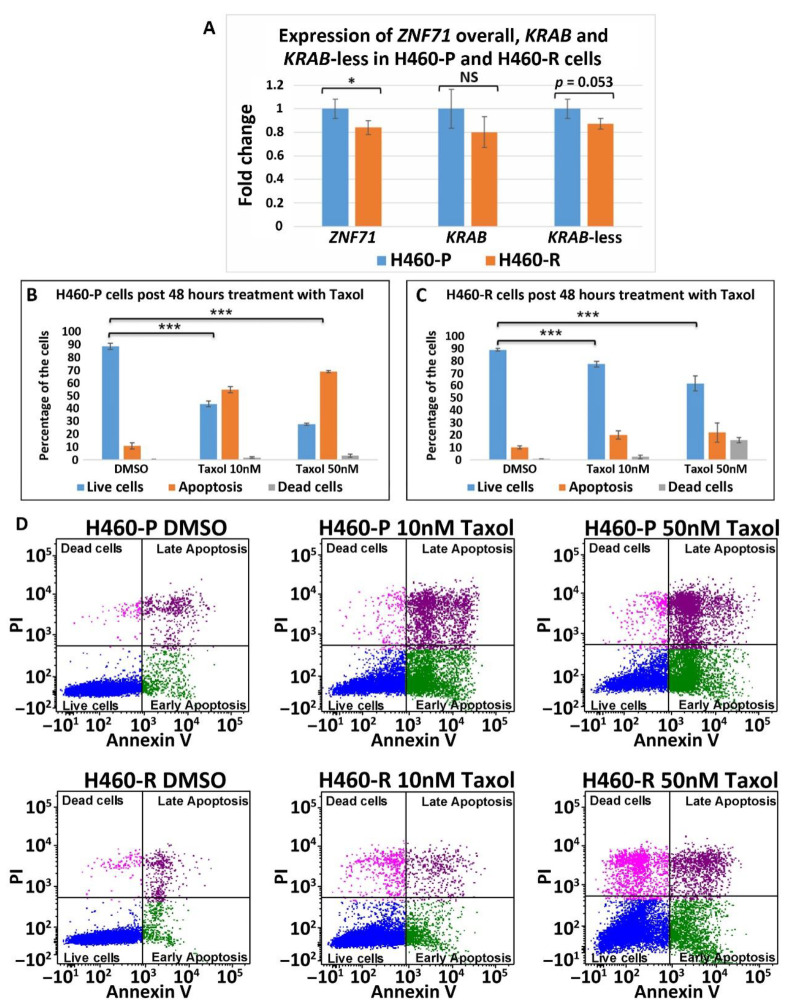
*ZNF71* isoforms in paclitaxel-resistant (H460-R) and parental (H460-P) cell lines. (**A**) Comparison of fold change of *ZNF71* overall, *KRAB*, and *KRAB*-less expression in H460-P and H460-R cell lines in qRT-PCR experiments. * *p* < 0.05, *** *p* < 0.001, NS: non-significance, two-sample *t*-tests. (**B**,**C**) H460-P (**B**) and H460-R (**C**) cells were treated either with DMSO or Taxol with the indicated doses. After 48 h, the cells were stained with Annexin V and PI and the percent of live/dead cells was determined by flow cytometry. Error bars represent standard deviation from three biological replicates. A mixed-effect model was used to assess the difference between two conditions, using R package *Ime4.* The *p*-value was calculated based on asymptotic *Z*-distribution. (**D**) Flow cytometry images for H460-P and H460-R, respectively, after treatment with either DMSO or Taxol.

## Data Availability

All links to publicly archived datasets are provided in the manuscript.

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
