# Peer review of "Molecular Analysis of ZNF71 KRAB in Non-Small-Cell Lung Cancer"

_ijms, 2021, doi:10.3390/ijms22073752_

Round 1

Reviewer 1 Report

This  study originates from publicly available RNA-seq datasets and analyzes the expression of ZNF71 overall as well as ZNF71 KRAB and ZNF71 KRAB-less isoforms in NSCLC. Main conclusions of the study are that: 1) both NSCLC patients’ tumor specimens (n=197) and multiple NSCLC cell lines (n=117) express higher levels of ZNF71 KRAB than ZNF71 KRAB-less isoform; 2) Patients with a higher ZNF71 KRAB expression had a significantly worse outcome than patients with a lower ZNF71 KRAB expression, while ZNF71 overall and KRAB-less expression was not prognostic in the same patient cohort; 3) in both NSCLC specimens and cell lines ZNF71 KRAB expression was associated with mRNA and protein expression of EMT markers; 4) ZNF71 KRAB was overexpressed in NSCLC cell lines resistant to docetaxel and paclitaxel treatment; and finally 5) the expression of ZNF71 KRAB isoform appears to be a more effective prognosticator than the expression of ZNF71 overall or KRAB-less. Overall, the presented results seem to support the  conclusions, however they also generate unanswered questions and matters of concerns that, at least in part, need to be addressed.

SPECIFIC COMMENTS

A more thorough comparison with the authors’ previously published results is necessary. Indeed, in reference 6 the authors showed that higher mRNA and protein expression of ZNF71 in NSCLC were associated with better prognosis, and that ZNF71 mRNA expression positively correlated with chemosensitivity to cisplatin, carboplatin, and taxol in NSCLC patients. How do the authors reconcile these previous results with the current data suggesting that ZNF71 KRAB expression is a negative prognostic marker in NSCLC, whereas  ZNF71 overall is not prognostic?

The authors understandably state that “ZNF71 KRAB isoform may provide added prognostic and predictive value to the 7-gene assay developed” in their previous study (ref. 6). Have the authors tried to replace overall ZNF71 with KRAB in their 7-gene prognostic signature, to see whether and how that may change the prognostic performance of the signature?

Introduction, line 74-5, “Furthermore, the association of ZNF71 isoforms and EMT and the implications in NSCLC prognosis were examined”. The Introduction is lacking a clear explanation for why an association between ZNF71 KRAB and EMT should be expected and worth being studied.

Also, a brief description of Figure 1 should be present in the Introduction. At present, one needs to read Materials and Methods to understand in detail the overall study scheme illustrated in the figure.

Results, line 98-100, “The expression of overall ZNF71 and its isoforms was not significantly different among different histological subtypes in patient tumors (ANOVA tests, Supplementary Figure 1). The expression of ZNF71 overall and KRAB isoform was significantly different among histological subtypes in NSCLC cell lines”:  what it is the clinical value of expression data in NSCLC cell lines, if they do not reflect those in patients’ tumors? It should be commented upon by the authors in the Discussion.

Fig. 3B and Suppl. Fig. 1 appear to be the same figure except that in the latter the expression of endogenous ZNF71 and the overexpression of ZNF1 in HEK-293T cells are shown. Thus, it’d seem logical to replace Fig. 3B with Suppl. Fig. 1 (and discard the current Fig. 3B).

Line 160-2, “In the Mesenchymal and the Low expression overlap phenotypes, there was no significant difference in patient survival between the high and low KRAB expression groups”: yet there is a non-significant trend also in these two phenotypes. This could be mentioned.

Line 184-5: The sentence “These more mesenchymal cell lines would NOT be comparable to patient tumors with the Epithelial phenotype, which showed a negative association of ZNF71 KRAB expression and patient survival” is not clear. Please, rephrase this part in a more decipherable manner.

Fig. 3A: the relative quantity of ZNF71 overall and KRAB in the SAEC normal lung cell line is significantly higher than in most of the NSCLC cell lines. It should be explained and discussed, as it casts some doubts and questions on the relevance of ZNF71 in lung carcinogenesis. Is ZNF71 downregulated in NSCLC as compared to normal lung tissue? Not knowing what is the pathogenetic role of ZNF71 in NSCLC (if any), one wonders based on these data whether it is oncogenic or tumor-suppressive.

The authors state that “the knock-down/knockout of ZNF71 did not significantly affect cell growth in NSCLC cell lines, implying that ZNF71 might not be involved in cell proliferation”. Can they provide at least a theoretical explanation for how ZNF71 overexpression might contribute to tumor progression? Is there any link to the postulated association between ZNF71 KRAB expression and EMT? These issues should be briefly discussed further by the authors.  

The section on the H460 cell line in the Results and Discussion is very elaborated and somehow confusing with respect to one of the take-home messages of the study, i.e. that ZNF71 KRAB overexpression may be associated with resistance to docetaxel and paclitaxel. The interpretation of ZNF71 overall expression being a “chemosensitivity marker in NSCLC tumors within the average response range to Taxol, which are neither intrinsically very resistant nor sensitive to paclitaxel (Taxol) treatment" is speculative and somehow cryptic. Perhaps, for the sake of manuscript clarity, this part should be removed, as it does not add anything essential for the main messages and complicates the interpretation of the text.

“Drug responses of 9 commonly used chemotherapeutic regimens in treating NSCLC were included in this study”: among these 9 drugs the authors mention etoposide, which is SOC drug used as 1st line against SCLC not NSCLC. Thus, the usage of etoposide in the study should be explained.

Moreover, the authors mention the EGFR-TKIs gefitinib and erlotinib, which are utilized to treat EGFR-mutated NSCLC as main indication. Were there EGFR-mutant cell lines in the used panel? The usage of these two EGFR-TKIs of 1st generation should be explained too.

“This categorization corresponds to the RECIST 1.1 system (i.e., complete response, partial response, and disease progression)”: RECIST includes also SD.

References: They are not written according to IJMS's instructions. Moreover, they should be aligned with the reference numbers. Finally, references 48, 50, 54, 55 are incomplete (missing issue and page numbers).

Reviewer 2 Report

This is a well-designed study on the role of ZNF71 KRAB expression in terms of prognosis and prediction to chemotherapy.

I have just a few criticisms:, as follows:

  1. Since it is well known that histology may have a role in adopting various chemotherapeutic regimens (e.g., pemetrexed or gem or tax in association with platinum), the authors at least should better define if the term NSCLC refers to non-squamous or squamous carcinoma (including cell line differentiation).
  2. at the same time, more details on the chemotherapeutic regimens better responding to ZNF71 KRAB setup need to be inserted in the results and discussion sections

Reviewer 3 Report

The manuscript presented by Guo and co-workers investigates the role of ZNF71 isoforms in NSCLC  cell lines and patients. Overall, the manuscript is well organized and written. Despite some minor corrections I have not any concern on its publication.

Minor concerns are :

  1. Row 106-107: "…using a cutoff of KRAB expression level measured with  transcripts per million (TPM) of 1.5,…" Please, describe how the cut-off was defined.
  2. Row 245, Figure 5: the figure title and the organization of the figure should be similar to the previous ones. The font of the  title of the figure as well as of y axis are too large.
  3. Supplementary file includes most patients data.  I suggest moving the entire supplementary file into manuscript appendices.

Round 2

Reviewer 1 Report

The authors deserve lots of credit for having addressed properly most of the previous comments. Their manuscript contains an impressive amount of data paving the way for interesting future studies and deserving publication. However, there remains some minor issues, especially concerning the presentation of data, that should be considered before complete acceptance.

SPECIFIC POINTS

Figure 3B: Add in the legend that the figure, in addition to WB of EMT markers, also shows the endogenous ZNF71 and the overexpression of ZNF1 in HEK-293T cells (top lanes). Moreover: is it correct to label the lanes in the figure as "Epithelial markers" or should it be "EMT markers" as mentioned in the figure legend?

Figure legend to Figure A2, “Adenocarcinoma (AC) had the highest ZNF71 overall expression”: based on the figure, it seems to be LCC not AC.

Introduction, Materials & Methods, and Results (paragraph 2.4): The reason for including etoposide among the "drugs commonly used to treat NSCLC" in the study is still not clear. The authors state that due to synergism and successful results of combining etoposide with cisplatin in SCLC, the drug was also used  to treat NSCLC, but they refer to a publication from 1992. Currently, etoposide is not indicated by international guidelines in the treatment of NSCLC.  Thus, the usage should be further justified, possibly by using more recent references in which the drug has been utilized against NSCLC, at least to show its applicability to this lung cancer type.

Table  A5 and A6: the specific EGFR mutations in the NSCLC cell lines need to be reported, as it is not clear where the mutations named in the two tables are in the EGFR gene. The clinically relevant mutations are well known to be in the TKD of EGFR (Exon 18-21) and, importantly, different missense mutations or insdels in these exons can have very different impact on the response to EGFR-TKIs of different generation. For ex. the missense mutation p.L858R is very sensitive to EGFR-TKIs of all 3 generations, while p.T790M is resistant to EGFR-TKIs of 1st-2nd generation, but responds to Osimertinib. Similarly, EGFR ex19insdels are usually responsive to EGFR-TKIs of all three generations, whereas most of EGFR ex20ins are not and require ex20ins-specific inhibitors.

Discussion, line 451-2, “different among histological subtypes in NSCLC epithelial cell lines”:  "epithelial" should be eliminated from the sentence, as the utilized NSCLC cell lines are epithelial (derived from lung carcinomas, that by definition are epithelial tumors).

Line 4524, “The association between ZNF71 overall and KRAB expression and stromal infiltration in patient tumors (Table A4) could possibly contribute to the observed expression discrepancy among NSCLC histological subtypes in cell lines and patient tumors”: Intuitively, this should be due to the fact that stromal infiltration is present in patient tumors, not in cell lines in vitro, but perhaps for the sake of clarity the authors should specify that more firmly in the sentence. In other words, does the association between ZNF1 overall and KRAB expression with stromal infiltration dilute the association with histologic type that is detectable in the NSCLC cell lines?

Methods, line 583: stable disease should be mentioned before disease progression, as it is indicative of at least some disease control by a treatment as compared to progression.

Check some spelling errors, like “sensitive”, “survivial” and possibly just a few others.
